# Facile Synthesis of Some Coumarin Derivatives and Their Cytotoxicity through VEGFR2 and Topoisomerase II Inhibition

**DOI:** 10.3390/molecules27238279

**Published:** 2022-11-28

**Authors:** Mohamed S. Gomaa, Ibrahim A. I. Ali, Gaber El Enany, El Sayed H. El Ashry, Samir M. El Rayes, Walid Fathalla, Abdulghany H. A. Ahmed, Samar A. Abubshait, Haya A. Abubshait, Mohamed S. Nafie

**Affiliations:** 1Department of Pharmaceutical Chemistry, College of Clinical Pharmacy, Imam Abdulrahman Bin Faisal University, P.O. Box 1982, Dammam 31441, Saudi Arabia; 2Department of Chemistry, Faculty of Science, Suez Canal University, Ismailia 41522, Egypt; 3Department of Physics, College of Science and Arts in Uglat Asugour, Qassim University, Buraidah 52571, Saudi Arabia; 4Scientific Department, Faculty of Engineering, Port Said University, Port Said 42526, Egypt; 5Chemistry Department, Faculty of Science, University of Alexandria, Alexandria 21526, Egypt; 6Chemistry Department, Faculty of Medicinal Science, University of Science and Technology, Aden 15201, Yemen; 7Chemistry Department, College of Science, Imam Abdulrahman Bin Faisal University, P.O. Box 1982, Dammam 31441, Saudi Arabia; 8Basic and Applied Scientific Research Center, Imam Abdulrahman Bin Faisal University, P.O. Box 1982, Dammam 31441, Saudi Arabia; 9Basic Science Department, Deanship of Preparatory Year and Supporting Studies, Imam Abdulrahman Bin Faisal University, P.O. Box 1982, Dammam 31441, Saudi Arabia

**Keywords:** amino acids, coumarin, DCC coupling, dipeptides, VEGFR2, topoisomerase II, docking studies

## Abstract

Novel semisynthetic coumarin derivatives were synthesized to be developed as chemotherapeutic anticancer agents through topoisomerase II, VEGFR2 inhibition that leads to apoptotic cancer cell death. The coumarin amino acids and dipeptides derivatives were prepared by the reaction of coumarin-3-carboxylic acid with amino acid methyl esters following the *N*,*N*-dicyclohexylcarbodiimide (DCC) method and 1-hydroxy-benzotriazole (HOBt), as coupling reagents. The synthesized compounds were screened towards VEGFR2, and topoisomerase IIα proteins to highlight their binding affinities and virtual mechanism of binding. Interestingly, compounds **4k** (Tyr) and **6c** (β-Ala-L-Met) shared the activity towards the three proteins by forming the same interactions with the key amino acids, such as the co-crystallized ligands. Both compounds **4k** and **6c** exhibited potent cytotoxic activities against MCF-7 cells with IC_50_ values of 4.98 and 5.85 µM, respectively causing cell death by 97.82 and 97.35%, respectively. Validating the molecular docking studies, both compounds demonstrated promising VEGFR-2 inhibition with IC_50_ values of 23.6 and 34.2 µM, compared to Sorafenib (30 µM) and topoisomerase-II inhibition with IC_50_ values of 4.1 and 8.6 µM compared to Doxorubicin (9.65 µM). Hence, these two promising compounds could be further tested as effective and selective target-oriented active agents against cancer.

## 1. Introduction

Cancer is the leading or second leading cause of death from noncommunicable diseases in the world [1]. Cancer is a multifaceted disease characterized by the growth of abnormal cells that are out of control. Tumor formation and development are linked to apoptosis, which is thought to be a highly programmed mechanism of cell death. One of the main reasons for their prominence was their wide range of pharmacological actions. The design of coumarin skeleton substitutions has a significant impact on the therapeutic applications of these compounds [2].

Anticancer, cytotoxic, and antiproliferative activity were among the many biological effects shown by coumarin derivatives. It’s not just that they work well against cancer, but that they rarely cause side effects [3]. Herbal medicines containing coumarins were traditionally derived from various plant, fungal, and bacterial sources [4,5]. Coumarin is frequently associated with biological activity, such as anti-cancer [6], antifungal [7], anti-HIV [8,9], inhibit lipid peroxidation [10] and anti-clotting. For many years, the therapy of angina pectoris has included the use of carbochromen, a powerful selective coronary vasodilator [11]. Seseline has been shown to have anti-HIV effects [12,13], while novobiocin [14] has antibiotic properties and wedelolactone [15] is used as venomous snakebite antidote Figure 1A.

Coumarins are a class of natural compounds commonly found in a variety of plant families [16], as naturally occurring antitumor compounds. Hyaluronic acid, a key component in tumor growth and progression, is inhibited by 4-methylumbelliferone [17]. Cancers of the pancreas, kidney, prostate, ovaries, and breasts are all effectively targeted by this chemopreventive and chemotherapeutic drug [18]. Many different semi-synthetic coumarin compounds, with varying substitution patterns, have been identified as powerful anticancer drugs, as shown in Figure 1B, as examples of some anticancer coumarin-based derivatives (**I**–**VI**) [19,20,21,22].

Recently, studies have demonstrated that coumarin-based compounds can inhibit immune regulation, cell growth and differentiation, and cell differentiation through VEGFR2 [23,24,25], and topoisomerase II [26,27,28] inhibition, which have been considered attractive targets for the development of anticancer agents. Almost all cellular responses to VEGF are mediated by VEGFR-2, a type III transmembrane receptor tyrosine kinase (RTK) present in a wide variety of cancer cells. It is considered as the most important transducer of VEGF-dependent apoptosis and angiogenesis [29]. Hence, the VEGFR-2 inhibitory signaling pathway has become a crucial strategy for the identification and development of novel therapeutics for a variety of human malignancies [30].

N.S. Reddy et al. [31] have outlined the growth inhibitory properties of the novel coumarin sulfonamide. These coumarin derivatives activated c-Jun N-terminal kinase 1 (JNK 1) [32]. Reddy et al. have designed and synthesized a series of novel coumarin-3-carboxamides and examined the antioxidant, anti-inflammatory and against anti-human immunodeficiency activities [33,34]. For a long time, organic and medicinal chemists have been interested in the synthesis of coumarins and their derivatives. Structure modification on coumarins focused on the attachment of pharmacophoric groups at C-3, C-4 and C-7 positions for biological evaluation, including anti-microbial, anti-HIV, anti-cancer, lipid-lowering, anti-oxidant, and anti-coagulation activities [35]. Considering the highly valuable biological and pharmaceutical properties of coumarins, it was aimed to synthesize a series of coumarins containing a variety of pharmacophores, amino acids and peptides at position 3, and to investigate the hit compounds through the molecular docking screening as cytotoxic against HepG2, and MCF-7 cancer cells.

## 2. Results and Discussion

### 2.1. Chemistry

In continuation of our efforts in synthesizing various bioactive molecules [36], it was found desirable to synthesize a series of coumarin-3-amino acid methyl esters and dipeptides. By interacting with the receptor recognition sites, these compounds were designed to enhance the binding affinity of the coumarin moiety.

Condensation of diethyl malonate with salicylaldehyde afforded the coumarin-3-carboxylic acid ethyl ester, which upon hydrolysis gave coumarin-3-carboxylic acid (**3**, Figure 1) [32].

The acid **3** is an excellent precursor for the structure modification via peptide coupling. Peptide bonds were first introduced in the literature using a variety of coupling reagents, which react carboxylic acid with amino acid methyl ester [37]. A convenient coupling method [38] was employed for the formation of peptides by reaction of the carboxylic acid group with amino acid esters, using *N*,*N*-dicyclohexylcarbodiimide (DCC) and 1-hydroxy-benzotriazole (HOBt), as coupling reagents. With its ability to reduce racemization during carbodiimide peptide coupling, HOBt is a common method [39].

Thus, the treatment of coumarin-3-carboxylic acid (**3**) with amino acid methyl esters (glycine, L-alanine, L-leucine, L-serine, L-phenylglycine, β-alanine, L-valine, L-methionine, L-tryptophan, L-tyrosine) in the presence of coupling reagents, DCC and HOBT, afforded the amino ester derivatives **4a**–**l**, respectively, in the 63–87% yield, Figure 2.

Saponification of esters **4c**, **4e** and **4i** (β-Ala, L-Val, and L Met) in alcoholic KOH afforded the free acid derivatives **5c**, **5e** and **5i** (β-Ala, L-Val, and L Met) in 58–82% yield, Figure 3. Next, our target was the formation of dipeptide derivatives following the above procedure by the treatment of acid derivatives **5c**, **5e** and **5i** (β-Ala, L-Val, and L Met) with amino acid esters, via the DCC coupling method to give the dipeptides **6**–**8**(**a**–**c**) in 52–69% yield, Figure 3.

The structure assignment of the amino acid ester **4a**–**l** and dipeptide **6**–**8**(**a**–**c**) based on spectral and physicochemical analysis; Figure 2. All the isolated products exhibited a rather interesting, fixed conformation, as represented in Figure 2, and indicated from all ^1^H NMR spectra. Thus, the ^1^H NMR spectrum of 3-methyl-2-[(2-oxo-2*H*-chromene-3-carbonyl)-amino]-butyric acid methyl ester **4e** (L-Val) demonstrated an interesting exchangeable singlet down fielded signal at δ 9.24 ppm corresponding to the NH group participating in an intramolecular hydrogen bond interaction of the type N–H···O=C [40]. The ^1^H NMR spectrum of **4e** (L-Val) also demonstrated signals at δ 1.01, 3.75, 4.65 and 8.87 corresponding to 2CH_3_, OCH_3_, CH_(Val)_ and CH_(coumarin)_, respectively.

The ^1^H NMR spectrum of methyl 3-{3-methyl-2-[(2-oxo-2*H*-chromene-3-carbonyl)-amino]-butyryl-amino} propanoate (**7a**) (L-Val-β-Ala) similarly demonstrated signals at δ 9.20, 8.87, 6.60, 4.35, 3.59–3.47, 2.53 and 3.65 ppm corresponding NH, CH_(coumarin)_, NH, CH_(Val)_, 2CH_2_ and OCH_3_ groups, respectively; Figure 2. The ^13^C NMR spectrum of (**7a**) (L-Val-β-Ala) reveal signals at δ 171.4, 170.2, 167.3, 166.4, 61.8, 56.9, 34.3, 38.7, 17.5, and 17.3 ppm corresponding to four C=O, CH, OCH_3_, CH_2_, CH_2_, CH_3_ and CH_3_ groups, respectively.

### 2.2. In Silico Studies

#### 2.2.1. Molecular Docking

The synthesized derivatives were screened for their targets using a molecular docking study. They were screened towards VEGFR-2, and topoisomerase IIα, based on the previous studies that highlighted their binding affinities. As seen in Table 1, the compounds/protein interactions were summarized as a result of docking analysis.

Regarding the molecular docking study towards VEGFR2, the co-crystallized ligand of VEGFR2 protein formed three HB interactions with Asp 1064, Cys 919 and Lys 868 as the key amino acids. Interestingly, most compounds formed one important interaction like the native ligand. Additionally, for the molecular docking study towards topoisomerase IIα, the co-crystallized ligand of Topoisomerase IIα protein formed one HB with Ser 149 as the key amino acid. Thus, both compound **4k** and **6c** structures represents the important pharmacophoric regions that made them share the activity towards the three proteins by forming the same interactions with the key amino acids, such as the co-crystallized ligands. As observed in Figure 3 with the surface and interactive representations, compound **4k** maintained the binding disposition of the co-crystallized ligand of VEGFR2 (**A**), and Topoisomerase II (**B**).

#### 2.2.2. ADME Pharmacokinetics

In order to determine the physicochemical and drug-like properties of compounds **4k** and **6c** towards the examined proteins, bioinformatics study was performed. All of the tested chemicals were readily permeable to and absorbed. As shown in Table 2, typically, compounds contained 2 H-bonding donors and 6 to 7 H-bonding acceptors. In addition, they exhibited log *p* values (1.81–2.69), so they were well tolerated by cell membranes. The ratable bond number (nrotb) should be less than 10 for effective regulation of conformational changes and oral bioavailability. In addition, the BOILED-Egg model found that the compounds had good absorption in the gastrointestinal tract, as shown in Figure 4.

### 2.3. Biological Assessment

#### 2.3.1. Cytotoxicity against MCF-7 and HepG2 Cancer Cell Lines

Based on the preliminary of the molecular docking studies of all tested compounds, two hit compounds **4k** and **6c** with the highest binding affinities towards the tested targets to be screened as cytotoxic against MCF-7 and HepG2 cell lines. As seen in Figure 5, compounds **4k** and **6c** exhibited promising cytotoxicity against MCF-7 cells with IC_50_ values of 4.98 and 5.85 µM, respectively, causing cell death by 97.82 and 97.35%, respectively. Additionally, compound **4k** exhibited promising cytotoxicity against HepG2 cells with an IC_50_ value of 9.4 µM, while compound **6c** showed moderate cytotoxicity with an IC_50_ value of 33.88 µM. Hence, these compounds exhibited promise cytotoxic activity.

#### 2.3.2. Enzyme Target Activity

Based on the primarily molecular docking studies of the highest binding affinities of **4k** and **6c**, enzymatic target activities of both compounds towards VEGFR2, topoisomerase II proteins were tested. As summarized in Table 3, compounds **4k** and **6c** demonstrated promising VEGFR-2 inhibition with IC_50_ values of 23.6 and 34.2 µM, compared to Sorafenib (30 µM) and topoisomerase-II inhibition with IC_50_ values of 4.1 and 8.6 µM compared to Doxorubicin (9.65 µM). Our results of VEGFR2 inhibition agreed with previous studies [23,24] that exhibited promising VEGFR2 inhibition of newly synthesized coumarin derivatives aligned with cytotoxicity in breast cancer cells with promising IC_50_ values. Other studies [26,41,42] designed some novel coumarin derivatives as cytotoxic activities through topoisomerase inhibition in some cancer cell lines including MCF-7 cells. There were previous studies [43,44] linking apoptosis as the mechanistic cell death upon treatment to VEGFR2 and topoisomerase inhibition, that are emerged as a prime approach for discovering new therapies for many apoptosis-dependent anticancer drugs.

## 3. Experimental Methods

### 3.1. Synthesis

Solvent was purified and dried in the usual way. The boiling range of the petroleum ether used was 40–60 °C. Thin layer chromatography (TLC): silica gel 60 F_254_ plastic plates (E. Merck, layer thickness 0.2 mm) detected by UV absorption. Melting points were determined on a Buchi 510 melting-point apparatus and the values are uncorrected. NMR spectra were measured with Bruker (300 MHz) (Waltham, MA, USA). Tetramethylsilane (TMS, 0.00 ppm) was used as the internal standard. The mass spectra were recorded on a Finnigan (MAT312) and Jeol (JMS.600H); HRMS were recorded with Thermo Finnegan (MAT 95XP) (San Jose, CA, USA).

General procedure for the preparation of coumarin-3-amino acid methyl esters (**4a–l**)**:** To a cold solution (−5 °C) of the amino acid methyl ester hydrochloride (2.0 mmol) in acetonitrile (20 mL) containing triethyl amine (0.28 mL, 2.0 mmol), coumarin-3-carboxylic acid (**3**) (0.38 g, 2.0 mmol), dicyclohexylcarbodiimide (DCC) (0.414 g, 2.0 mmol) and 1-hydroxybenzotriazole (HOBT) (0.27 g, 2.0 mmol) were added successively. The reaction mixture was stirred at 0 °C for one hour, at 5 °C for one hour, and then at room temperature for 12 h. The precipitated dicyclohexylurea was filtered off and the filtrate was evaporated under reduced pressure. The residue was dissolved in ethyl acetate, filtered and the filtrate was washed with 5% NaHCO_3_, 1N HCl and saturated solution of sodium chloride then dried over anhydrous sodium sulphate. After evaporation of the solvent, the remaining oily residue was triturated with petroleum ether (b.p. 40–60 °C) at 0 °C with scratching. The solid residue was filtered off and crystallized from petroleum ether/ethyl acetate. Charcterization analyses including NMR and MS for the synthesized compounds were supported as supplementry.

Methyl [(2-oxo-2*H*-chromene-3-carbonyl)-amino] ethanoate (**4a**) From glycine methyl ester hydrochloride, yield, 0.45 g (87%); mp 173–175 °C. ^1^H NMR (300 MHz, CDCl_3_): δppm: 3.77 (s, 3H, OCH_3_), 4.24 (d, *J* = 5.6 Hz, 2H, CH_2_), 7.34–7.41 (m, 2H, Ar-H), 7.63–7.69 (m, 2H, Ar-H), 8.89 (s, 1H, CH), 9.22 (br s, 1H, NH); HR EIMS: Calcd for C_13_H_11_NO_5_ (261.2301): Found (261.2110).

Methyl 2-[(2-oxo-2*H*-chromene-3-carbonyl)-amino] propanoate (**4b**)**.** From L-alanine methyl ester hydrochloride, yield, 0.46 g (84%); mp 104–106 °C. ^1^H NMR (300 MHz, CDCl_3_): δppm: 1.52 (d, *J* = 7.1 Hz, 3H, CH_3_), 3.76 (s, 3H, OCH_3_), 4.73 (m, 1H, CH), 7.33–7.40 (m, 2H, Ar-H), 7.62–7.68 (m, 2H, Ar-H), 8.86 (s, 1H, CH), 9.22 (d, *J* = 6.5 Hz, 1H, NH); HR EIMS: Calcd for C_14_H_13_NO_5_ (275.2567): Found (275.2390).

Methyl 3-[(2-oxo-2*H*-chromene-3-carbonyl)-amino] propanoate (**4c**). From β-alanine methyl ester hydrochloride, yield, 0.44 g (81%); mp 135–137 °C. ^1^H NMR (300 MHz, CDCl_3_): δppm: 2.65 (t, *J* = 6.4 Hz, 2H, CH_2_), 3.70–3.76 (m, 5H, CH_2_, OCH_3_), 7.35–7.40 (m, 2H, Ar-H),7.64–7.68 (m, 2H, Ar-H), 8.87 (s, 1H, CH), 9.11 (bs, 1H, NH); HR EIMS: Calcd for C_14_H_13_NO_5_ (275.2567): Found (275.1971).

Methyl 3-hydroxy-2-[(2-oxo-2*H*-chromene-3-carbonyl)-amino]-propanoate (**4d**)**.** From L-serine methyl ester hydrochloride, yield, 0.44 g (76%); mp 177–179 °C. ^1^H NMR (300 MHz, CDCl_3_): δppm: 1.66 (bs, D_2_O Exchangeable, 1H, OH), 3.81 (s, 3H, OCH_3_), 4.02–4.11 (m, 2H, CH_2_), 4.82–4.87 (m, 1H, CH), 7.34–7.41 (m, 2H, Ar-H), 7.64–7.69 (m, 2H, Ar-H), 8.88 (s, 1H, CH), 9.56 (d, *J* = 6.9 Hz, 1H, NH); HR EIMS: Calcd for C_14_H_13_NO_6_ (291.2561): Found (291.2110).

Methyl 3-methyl-2-[(2-oxo-2*H*-chromene-3-carbonyl)-amino] butanoate (**4e**)**.** From L-valine methyl ester hydrochloride, yield, 0.42 g (69%); mp 75–77 °C. ^1^H NMR (300 MHz, CDCl_3_): δppm: 1.01 (d, *J* = 6.0 Hz, 6H, 2CH_3_), 2.27–2.33 (m, 1H, CH), 3.75 (s, 3H, OCH_3_), 4.65 (dd, *J* = 5.3, 7.9 Hz, 1H, CH), 7.24–7.41 (m, 2H, Ar-H), 7.62–7.68 (m, 2H, Ar-H), 8.87 (s, 1H, CH), 9.24 (d, *J* = 7.9 Hz, 1H, NH); HR EIMS: Calcd for C_16_H_17_NO_5_ (303.1067): Found (303.1097).

Methyl 4-methyl-2-[(2-oxo-2*H*-chromene-3-carbonyl)-amino] pentan-oate (**4f**). From L-leucine methyl ester hydrochloride, yield, 0.40 g (63%); mp 95–97 °C. ^1^H NMR (300 MHz, CDCl_3_): δppm: 0.96 (2d, *J* = 3.1 Hz, 6H, 2CH_3_),1.72–1.75 (m, 3H, CH, CH_2_), 3.74 (s, 3H, OCH_3_), 4.73–4.76 (m, 1H, CH), 7.33–7.41 (m, 2H, Ar-H), 7.62–7.68 (m, 2H, Ar-H), 8.87 (s, 1H, CH), 9.10 (d, *J* = 7.4 Hz, 1H, NH); HR EIMS: Calcd for C_17_H_19_NO_5_ (317.1263): Found (317.1178).

Dimethyl 2-[(2-oxo-2*H*-chromene-3-carbonyl)-amino] succinate (**4g**)**.** From L-aspartic methyl ester hydrochloride, yield, 0.47 g (70%); mp 69–71 °C. ^1^H NMR (300 MHz, CDCl_3_): δppm: 2.87–2.91 (m, 2H, CH_2_), 3.66 (s, 3H, OCH_3_), 3.72 (s, 3H, OCH_3_), 4.87–5.09 (m, 1H, CH), 7.31–7.35 (m, 2H, Ar-H), 7.55–7.58 (m, 2H, Ar-H), 8.83 (s, 1H, CH), 9.11 (d, *J* = 7.9 Hz, 1H, NH). ^13^C NMR spectrum, (75.0 MHz, CDCl_3_), δ, ppm: 35.2, 51.6, 55.7, 56.9, 123.4, 125.5, 126.3, 127.4, 128.8, 130.2, 148.6, 149.8, 165.2 (C=O), 166.4 (C=O), 172.6 (C=O), 172.7 (C=O). HR EIMS: Calcd for C_16_H_15_NO_7_ (333.1077): Found (333.1087).

Dimethyl 2-[(2-oxo-2*H*-chromene-3-carbonyl)-amino] pentanedioate (**4h**)**.** From L- Glutamic methyl ester hydrochloride, yield, 0.48 g (69%); mp 64–66 °C. ^1^H NMR (300 MHz, CDCl_3_): δppm: 2.27–2.31 (m, 2H, CH_2_), 2.35–2.38 (m, 2H, CH_2_), 3.71 (s, 3H, OCH_3_), 3.75 (s, 3H, OCH_3_), 4.37–4.44 (m, 1H, CH), 7.33–7.36 (m, 2H, Ar-H), 7.49–7.52 (m, 2H, Ar-H), 8.80 (s, 1H, CH), 9.17 (d, *J* = 7.9 Hz, 1H, NH). ^13^C NMR spectrum, (75.0 MHz, CDCl_3_), δ, ppm: 24.4, 25.2, 53.6, 56.1, 57.3, 123.4, 125.1, 126.4, 127.7, 128.5, 130.6, 148.8, 149.7, 165.4 (C=O), 166.3 (C=O), 172.4 (C=O), 172.6 (C=O). HR EIMS: Calcd for C_17_H_17_NO_7_ (347.1065): Found (347.1077).

Methyl 4-methylsulfanyl-2-[(2-oxo-2H-chromene-3-carbonyl)-amino] butanoate (**4i**). From L-methionine methyl ester hydrochloride, yield, 0.46 g (68%); mp 81–83 °C. ^1^H NMR (300 MHz, CDCl_3_): δppm: 2.09 (s, 3H, SCH_3_). 2.12–2.29 (m, 2H, CH_2_), 2.55–2.60 (m, 2H, CH_2_), 3.77 (s, 3H, OCH_3_), 4.85–4.92 (m,1H, CH), 7.34–7.41 (m, 2H, Ar-H), 7.63–7.68 (m, 2H, Ar-H), 8.87 (s, 1H, CH), 9.25 (d, *J* = 7.5 Hz, 1H, NH). ^13^C NMR spectrum, (75.0 MHz, CDCl_3_), δ, ppm: 17.1, 29.5, 33.9, 54.9, 56.5 (OCH_3_), 123.2, 126.4, 127.1, 128.2, 128.6, 130.1, 148.5, 149.9, 164.8 (C=O), 169.2 (C=O), 171.4 (C=O). HR EIMS: Calcd for C_16_H_17_NO_5_S (335.0827): Found (335.0861).

Methyl [(2-oxo-2*H*-chromene-3-carbonyl)-amino]-phenylacetate (**4j**)**.** From DL-phenylglycine methyl ester hydrochloride, yield, 0.49 g (73%); mp 189–192 °C. ^1^H NMR (300 MHz, CDCl_3_): δppm: 3.75 (s, 3H, OCH_3_), 5.70 (d, *J* = 6.4 Hz, 2H, CH), 7.32–7.47 (m, 7H, Ar-H), 7.64–7.67 (m, 2H, Ar-H), 8.85 (s, 1H, CH), 9.73 (d, *J* = 6.4 Hz, 1H, NH). ^13^C NMR spectrum, (75.0 MHz, CDCl_3_), δ, ppm: 55.3, 56.6, 121.7, 125.3, 126.2, 127.7, 127.9, 128.8, 128.6, 129.3, 130.1, 135.4, 136.1, 148.5, 149.7, 166.6 (C=O), 167.2 (C=O), 172.4 (C=O). HR EIMS: Calcd for C_19_H_15_NO_5_ (337.0950): Found (337.0977).

Methyl 3-(4-hydroxyphenyl)-2-[(2-oxo-2*H*-chromene-3-carbonyl)-amino]-propanoate (**4k**). From L-tyrosine methyl ester hydrochloride, yield, 0.55 g (76%); mp 112–114 °C. ^1^H NMR (300 MHz, CDCl_3_): δppm: 2.98–3.23 (m, 2H, CH_2_), 3.73 (s, 3H, OCH_3_), 4.88–4.98 (m,1H, CH), 6.62–6.76 (m, 3H, Ar-H), 6.90 (bs, D_2_O Exchangeable, 1H, OH), 7.06–7.95 (m, 2H, Ar-H), 7.57–7.66 (m, 3H, Ar-H), 8.77 (s, 1H, CH), 9.29 (d, *J* = 7.4 Hz, 1H, NH). HR EIMS: Calcd for C_20_H_17_NO_6_ (367.0950): Found (367.0923).

Methyl 3-(1*H*-Indol-3-yl)-2-[(2-oxo-2*H*-chromene-3-carbonyl)-amino]-propanoate (**4l**)**.** From L-tryptophan methyl ester hydrochloride, yield, 0.49 g (64%); mp 138–139 °C. ^1^H NMR (300 MHz, CDCl_3_): δppm: 3.30–3.38 (m, 2H, CH_2_), 3.65 (s, 3H, OCH_3_), 4.85–4.89 (m,1H, CH), 6.93–7.07 (m, 2H, Ar-H), 7.10–7.23 (m, 2H, Ar-H), 7.33–7.49 (m, 3H, Ar-H), 10.97 (bs, 1H, NH), 7.69–7.98 (m, 2H, Ar-H), 8.89 (s, 1H, CH), 9.14 (d, *J* = 7.2 Hz, 1H, NH). ^13^C NMR spectrum, (75.0 MHz, CDCl_3_), δ, ppm: 34.1, 55.7, 56.9, 104.7, 117.7, 118.4, 120.3, 121.2, 125.4, 126.7, 127.1, 127.6, 128.2, 128.8, 130.3, 135.4, 136.2, 148.7, 149.9, 166.2, (C=O), 167.4 (C=O), 171.4 (C=O). HR EIMS: Calcd for C_22_H_18_N_2_O_5_ (390.1237): Found (390.1201).

General procedure for preparation of coumarin-3-amino acids (**5c, 5e** and **5i**)**:** A solution of **4c**, **4e** and **4i** (10.0 mmol) in ethanol (40 mL) and 5% KOH (15 mL) was stirred for 2–4h. The solution was evaporated to dryness and the residue was dissolved in water and acidified with dil HCl. The white precipitate was filtered, dried and crystallized from ethanol to give **5c, 5e** and **5i**.

3-[(2-Oxo-2*H*-chromene-3-carbonyl)-amino]-propanoic acid (**5c**)**:** yield, 0.42 g (82%); mp 195–197 °C. ^1^H NMR (300 MHz, CDCl_3_): δppm: 1.67 (bs, D_2_O Exchangeable,1H, OH), 2.72 (t, *J* = 6.2 Hz, 2H, CH_2_), 3.72–3.75 (m, 2H, CH_2_), 7.24–7.40 (m, 2H, Ar-H), 7.63–7.69 (m, 2H, Ar-H), 8.89 (s, 1H, CH), 9.16 (bs, 1H, NH). ^13^C NMR spectrum, (75.0 MHz, CDCl_3_), δ, ppm: 36.1, 36.9, 125.4, 126.7, 127.6, 128.2, 128.8, 130.3, 148.7, 149.9, 166.5 (C=O), 167.3 (C=O), 170.4 (C=O). HR EIMS: Calcd for C_13_H_11_NO_5_ (261.0637): Found (261.0704).

3-Methyl-2-[(2-oxo-2*H*-chromene-3-carbonyl)-amino] butanoic acid (**5e**) yield, 0.45 g (77%); mp 210–212 °C. ^1^H NMR (300 MHz, CDCl_3_): δppm: 1.06 (d, *J* = 6.8 Hz, 6H, 2CH_3_), 2.36–2.42 (m, 1H, CH), 3.43 (bs, D_2_O Exchangeable, 1H, OH), 4.65 (dd, *J* = 4.8, 7.8 Hz,1H, CH), 7.34–7.42 (m, 2H, Ar-H), 7.64–7.69 (m, 2H, Ar-H), 8.90 (s, 1H, CH), 9.30 (d, *J* = 7.8 Hz, 1H, NH). ^13^C NMR spectrum, (75.0 MHz, CDCl_3_), δ, ppm: 17.1, 17.3, 28.0, 63.7, 125.7, 126.3, 127.4, 128.6, 128.9, 130.7, 149.4, 149.8, 166.8 (C=O), 167.7(C=O), 177.4 (C=O). HR EIMS: Calcd for C_15_H_15_NO_5_ (289.0950): Found (289.1135).

4-Methylsulfanyl-2-[(2-oxo-2*H*-chromene-3-carbonyl)-amino]-butanoic acid (**5i**)**:** yield, 0.37 g (58%); mp 108–109 °C. ^1^H NMR (300 MHz, CDCl_3_): δppm: 2.11 (s, 3H, SCH_3_), 2.15–2.36 (m, 2H, CH_2_), 2.61–2.65 (m, 2H, CH_2_), 3.92 (bs, D_2_O Exchangeable, 1H, OH), 4.85–4.92 (m,1H, CH), 7.35–7.42 (m, 2H, Ar-H), 7.65–7.70 (m, 2H, Ar-H), 8.91 (s, 1H, CH), 9.31 (d, *J* = 6.8 Hz, 1H, NH). ^13^C NMR spectrum, (75.0 MHz, CDCl_3_), δ, ppm: 16.9, 29.4, 33.6, 55.3, 125.7, 126.6, 127.4, 128.6, 128.8, 130.4, 148.8, 149.7, 166.8 (C=O), 171.2 (C=O), 177.1 (C=O). HR EIMS: Calcd for C_15_H_15_NO_5_S (321.0671): Found (321.1127).

General procedure for preparation of coumarin-3-dipeptides methyl esters **6**–**8** (**a**–**c**). Dipeptides **6**–**8** (**a**–**c**) were prepared according to above method.

Methyl 3-{3-[(2-oxo-2*H*-chromene-3-carbonyl)-amino]-propionyl amino} propanoate (**6a**): (abbreviated: β-Ala-β-Ala) yield, 0.48 g (69%); mp 127–8 °C. ^1^H NMR (300 MHz, CDCl_3_): δppm: 2.49–2.55 (m, 4H, 2CH_2_),3.33–3.72 (m, 4H, 2CH_2_), 3.74 (s, 3H, OCH_3_), 6.55 (bs,1H, NH), 7.34–7.43 (m, 2H, Ar-H), 7.61–7.68 (m, 2H, Ar-H), 8.86 (s, 1H, CH), 9.11 (bs, 1H, NH). ^13^C NMR spectrum, (75.0 MHz, CDCl_3_), δ, ppm: 33.8, 34.9, 37.8, 38.4, 56.8 (OCH_3_), 125.3, 126.5, 127.4, 128.8, 128.4, 130.6, 148.5, 149.1, 166.2 (C=O), 167.6 (C=O), 170.3 (C=O), 171.4 (C=O). HR EIMS: Calcd for C_17_H_18_N_2_O_6_ (346.3346): Found (346.2521).

Methyl 3-Methyl-2-{3-[(2-oxo-2*H*-chromene-3-carbonyl)-amino]-propionyl-amino}-butanoate (**6b**)**:** (abbreviated: β-Ala-L-Val) yield, 0.53 g (71%); mp 101–103 °C. ^1^H NMR (300 MHz, CDCl_3_): δppm: 1.08 (d, *J* = 6.8 Hz, 6H, 2CH_3_), 2.42–2.49 (m, 2H, CH_2_), 2.99–3.06 (m, 1H, CH), 3.29–3.41 (m, 2H, CH_2_), 3.66 (s, 3H, OCH_3_), 4.40–4.46 (m, 1H, CH), 6.71 (bs,1H, NH), 7.26–7.33 (m, 2H, Ar-H), 7.52–7.57 (m, 2H, Ar-H), 8.63 (s, 1H, CH), 9.04 (bs, 1H, NH). ^13^C NMR spectrum, (75.0 MHz, CDCl_3_), δ, ppm: 17.1, 17.3, 28.0, 34.8, 38.6, 56.7 (OCH_3_), 61.3, 125.3, 126.2, 127.6, 128.4, 128.3, 130.2, 149.3, 149.9, 166.6 (C=O), 167.5 (C=O), 170.1 (C=O), 171.2 (C=O). HR EIMS: Calcd for C_19_H_22_N_2_O_6_ (374.3235): Found (374.3248).

Methyl 4-methylsulfanyl-2-{3-[(2-oxo-2*H*-chromene-3-carbonyl)-amino]-propionylamino}-butanoate (**6c**)**:** (abbreviated: β-Ala-L-Met) yield, 0.51 g (69%); mp 122–123 °C. ^1^H NMR (300 MHz, CDCl_3_): δppm: 2.12 (s, 3H, CH_3_), 2.33–2.37 (m, 2H, CH_2_), 2.42–2.46 (m, 2H, CH_2_), 2.49–2.53 (m, 2H, CH_2_), 3.08–3.14 (m, 2H, CH_2_), 3.65 (s, 3H, OCH_3_), 4.41–4.45 (m, 1H, CH), 6.66 (bs,1H, NH), 7.20–7.27 (m, 2H, Ar-H), 7.51–7.54 (m, 2H, Ar-H), 8.72 (s, 1H, CH), 9.10 (bs, 1H, NH). ^13^C NMR spectrum, (75.0 MHz, CDCl_3_), δ, ppm: 17.2, 28.7, 33.3, 34.9, 38.3, 54.2, 55.7 (OCH_3_), 125.7, 126.3, 127.0, 128.4, 128.6, 130.5, 149.2, 149.7, 166.1 (C=O), 167.7 (C=O), 170.2 (C=O), 171.6 (C=O). HR EIMS: Calcd for C_19_H_22_N_2_O_6_S (406.3125): Found (406.3133).

Methyl 3-{3-methyl-2-[(2-oxo-2*H*-chromene-3-carbonyl)-amino]-butyryl-amino}-propanoate (**7a**)**:** (abbreviated: L-Val-β-Ala) yield, 0.49 g (66%); mp 149–151 °C. ^1^H NMR (300 MHz, CDCl_3_): δppm: 0.98 (d, *J* = 6.8 Hz, 6H, 2CH_3_), 2.30–2.36 (m, 1H, CH), 2.53 (t, *J* = 5.3 Hz, 2H, CH_2_), 3.47–3.59 (m, 2H, CH_2_), 3.65 (s, 3H, OCH_3_), 4.35 (dd, *J* = 6.1, 7.9 Hz, 1H, CH), 6.60 (bs, 1H, NH), 7.34–7.41 (m, 2H, Ar-H), 7.63–7.69 (m, 2H, Ar-H), 8.87 (s, 1H, CH), 9.20 (d, *J* = 7.9 Hz, 1H, NH). ^13^C NMR spectrum, (75.0 MHz, CDCl_3_), δ, ppm: 17.3, 17.5, 28.2, 34.3, 38.7, 56.9 (OCH_3_), 61.8, 125.2, 126.1, 127.4, 128.2, 128.7, 130.5, 149.2, 149.8, 166.4 (C=O), 167.3 (C=O), 170.2 (C=O), 171.4 (C=O). HR EIMS: Calcd for C_19_H_22_N_2_O_6_ (374.1477): Found (374.1451).

Methyl 3-Methyl-2-{3-methyl-2-[(2-oxo-2*H*-chromene-3-carbonyl)-amino]-butyrylamino}-butanoate (**7b**)**:** (abbreviated: L-Val-L-Val) yield, 0.47 g (59%); mp 97–99 °C. ^1^H NMR (300 MHz, CDCl_3_): δppm: 0.91 (d, *J* = 7.1 Hz, 6H, 2CH_3_), 1.01 (d, *J* = 6.5 Hz, 6H, 2CH_3_), 2.13–2.19 (m, 1H, CH), 2.31–2.38 (m, 1H, CH), 3.70 (s, 3H, OCH_3_), 4.46 (dd, *J* = 6.4, 8.0 Hz, 1H, CH), 4.53 (dd, *J* = 4.8, 8.6 Hz, 1H, CH), 6.60 (d, *J* = 8.6 Hz, 1H, NH), 7.34–7.41 (m, 2H, Ar-H), 7.56–7.70 (m, 2H, Ar-H), 8.89 (s, 1H, CH), 9.25 (d, *J* = 8.0 Hz, 1H, NH). ^13^C NMR spectrum, (75.0 MHz, CDCl_3_), δ, ppm: 16.9, 17.0, 17.1, 17.3, 28.9, 28.9, 55.8 (OCH_3_), 61.9, 62.1, 125.5, 126.3, 127.1, 128.4, 128.9, 130.7, 149.4, 149.7, 166.8 (C=O), 167.4 (C=O), 170.5 (C=O), 171.7 (C=O). HR EIMS: Calcd for C_21_H_26_N_2_O_6_ (402.1791): Found (402.1829).

Methyl 2-{3-methyl-2-[(2-oxo-2H-chromene-3-carbonyl)-amino]-butyrylamino} -4-methylsulfanyl butanoate (**7c**)**:** (abbreviated: L-Val-L-Met) yield, 0.45 g (52%); mp 100–101 °C. ^1^H NMR (300 MHz, CDCl_3_): δppm: 1.02 (d, *J* = 5.7 Hz, 6H, 2CH_3_), 2.04 (s, 3H, SCH_3_), 2.12–2.22 (m, 2H, CH_2_), 2.33–2.39 (m, 1H, CH), 2.47–2.56 (m, 2H, CH_2_), 3.73 (s, 3H, OCH_3_), 4.42 (dd, *J* = 6.3, 7.9 Hz, 1H, CH), 4.67–4.74 (m, 1H, CH), 6.76 (d, *J* = 7.7 Hz,1H, NH), 7.34–7.42 (m, 2H, Ar-H), 7.64–7.69 (m, 2H, Ar-H), 8.88 (s, 1H, CH), 9.23 (d, *J* = 7.9 Hz, 1H, NH). ^13^C NMR spectrum, (75.0 MHz, CDCl_3_), δ, ppm: 17.0, 17.1, 17.3, 28.9, 31.3, 33.9, 54.1, 56.2 (OCH_3_), 61.7, 125.2, 126.1, 127.3, 128.5, 128.8, 130.4, 149.1, 149.9, 166.5, (C=O), 167.6 (C=O), 170.1 (C=O), 171.4 (C=O). HR EIMS: Calcd for C_21_H_26_N_2_O_6_S (434.1512): Found (434.1518).

Methyl 3-{4-methylsulfanyl-2-[(2-oxo-2*H*-chromene-3-carbonyl)-amino]-butyrylamino}-propanoate (**8a**)**:** (abbreviated: L-Met-β-Ala) yield, 0.51 g (69%); mp 104–105 °C. ^1^H NMR (300 MHz, CDCl_3_): δppm: 2.17 (s, 3H, CH_3_), 2.21–2.25 (m, 2H, CH_2_), 2.40–2.44 (m, 2H, CH_2_), 2.43–2.48 (m, 2H, CH_2_), 3.11–3.15 (m, 2H, CH_2_), 3.71 (s, 3H, OCH_3_), 4.48–4.52 (m, 1H, CH), 6.71 (bs,1H, NH), 7.27–7.31 (m, 2H, Ar-H), 7.54–7.58 (m, 2H, Ar-H), 8.68 (s, 1H, CH), 9.07 (bs, 1H, NH). ^13^C NMR spectrum, (75.0 MHz, CDCl_3_), δ, ppm: 16.9, 27.4, 30.1, 33.6, 38.4, 54.3, 57.1 (OCH_3_), 125.2, 126.4, 127.3, 128.7, 128.6, 130.3, 148.7, 149.5, 166.6 (C=O), 167.2 (C=O), 170.1 (C=O), 171.0 (C=O). HR EIMS: Calcd for C_19_H_22_N_2_O_6_S (406.3244): Found (406.3252).

Methyl 3-methyl-2-{4-methylsulfanyl-2-[(2-oxo-2*H*-chromene-3-carbonyl)-amino]-butyrylamino}-butanoate **(8b):** (abbreviated: L-Met-L-Val) yield, 0.56 g (64%); mp 98–99 °C. ^1^H NMR (300 MHz, CDCl_3_): δppm: 0.92 (d, *J* = 7.1 Hz, 6H, 2CH_3_), 2.12 (s, 3H, SCH_3_), 2.15–2.25 (m, 3H, CH, CH_2_), 2.63–2.68 (m, 2H, CH_2_), 3.72 (s, 3H, OCH_3_), 4.52 (dd, *J* = 4.7, 7.5 Hz, 1H, CH), 4.82 (m, 1H, CH), 6.86 (d, *J* = 7.5 Hz,1H, NH), 7.35–7.42 (m, 2H, Ar-H), 7.64–7.69 (m, 2H, Ar-H), 8.89 (s, 1H, CH), 9.26 (d, *J* = 7.4 Hz, 1H, NH). ^13^C NMR spectrum, (75.0 MHz, CDCl_3_), δ, ppm: 17.1, 17.3, 17.6, 28.8, 33.5, 33.9, 54.4, 56.2 (OCH_3_), 61.7, 125.0, 126.5, 127.4, 128.3, 128.7, 130.1, 148.9, 149.8, 166.3 (C=O), 167.7 (C=O), 170.4 (C=O), 171.2 (C=O). HR EIMS: Calcd for C_21_H_26_N_2_O_6_S (434.1512): Found (434.1518).

Methyl 4-methylsulfanyl-2-{4-methylsulfanyl-2-[(2-oxo-2*H*-chromene-3-carbonyl)-amino]-butyrylamino}-butanoate (**8c**)**:** (abbreviated: L-Met-L-Met) yield, 0.45 g (52%); mp 109–111 °C. ^1^H NMR (300 MHz, CDCl_3_): δppm: 2.03 (s, 3H, SCH_3_), 2.08 (s, 3H, SCH_3_), 2.12–2.22 (m, 2H, CH_2_), 2.33–2.39 (m, 2H, CH_2_), 2.41–2.44 (m, 2H, CH_2_), 2.47–2.56 (m, 2H, CH_2_), 3.70 (s, 3H, OCH_3_), 4.43–4.49 (m, 1H, CH), 4.61–4.66 (m, 1H, CH), 6.74 (d, *J* = 7.7 Hz,1H, NH), 7.32–7.37 (m, 2H, Ar-H), 7.61–7.67 (m, 2H, Ar-H), 8.83 (s, 1H, CH), 9.15 (d, *J* = 7.9 Hz, 1H, NH). ^13^C NMR spectrum, (75.0 MHz, CDCl_3_), δ, ppm: 17.1, 17.4, 28.7, 28.6, 33.5, 33.9, 54.4, 54.7, 66.2 (OCH_3_), 125.0, 126.5, 127.4, 128.3, 128.7, 130.1, 148.9, 149.8, 166.3 (C=O), 167.7 (C=O), 170.4 (C=O), 171.2 (C=O). HR EIMS: Calcd for C_21_H_26_N_2_O_6_S_2_ (466.2611):Found (466.2656).

### 3.2. In Silico Studies

#### 3.2.1. Molecular Docking

The synthesized compounds were screened for binding to demonstrate possible interactions towards some molecular targets of VEGFR2 (PDB = 3U6J), and Topoisomerase IIα (PDB = 1ZXM) using AutoDock Vina as the docking software. In each protein, water molecules and ions were removed from each receptor (protein) using Maestro. In addition, hydrogens were incorporated into the protein during the optimization step. Amino acid residues with missing atoms and those with alternate positions for some of their atoms were optimized. All synthesized derivatives were chemically and energetically optimized according to Nafie et al., 2019 [45,46]. The grid-box size (center_x_y_z: –24.11, −13.01, and 3.64; spacing dimensions of the grid were of 1 Å) was prepared, AutoDock Vina was used for molecular docking for binding poses generation and docking score calculations [47], and Finally, the docked chemical and protein interaction with the key amino acids was analysed using Chimera, a visualisation software.

#### 3.2.2. ADME Pharmacokinetics

In silico ADME pharmacokinetics parameters of the lead compounds were calculated using online websources including “MolSoft”, “Molinspiration”, and “SwissADME”, with their websites, as previously described [48].

### 3.3. Biological Assessment

#### 3.3.1. Cytotoxicity Using MTT Assay

Breast cancer (MCF-7) and liver cancer (HepG2) cell lines were collected from National Cancer Institute in Cairo and grown in “RPMI-1640 medium L-Glutamine. The cells were grown in 10% fetal bovine serum (FBS) and 1% penicillin-streptomycin using standard tissue culture protocol. On the second day, cells were grown in triplicate on a 96-well plate at a density of 5 × 10^4^ cells and incubated with the investigated compounds at “0.01, 0.1, 1, 10, and 100 µM”. Cell viabilioty was measured with MTT solution. The absorbance was measured using an ELISA microplate reader. After comparing the treated groups to the control, the IC_50_ values were recorded to indicate the level of viability following previous work [49,50,51].

#### 3.3.2. Enzyme Target Assays

VEGFR2 (KDR) kinase assay kit “BPS Bioscience Corporation catalog#40325” and Topo isomearse II screening kit “TopoGEN, Inc., Columbus, OH, USA” were utilized to invetstiage the enzyme activity for the most active compounds following the manufacturer instructions. Inhibition of autophosphorylation by tested compounds was determined. The IC_50_ values were determined using GraphPad Prism 7 software from curves depicting the percentage of inhibition using eight concentrations of each compound [43,44].

## 4. Conclusions

Herein, a novel series of coumarin-based amino acid and dipeptide derivatives synthesized as cytotoxic agents through VEGFR2 and Topoisomerase II inhibition. The coumarin amino acids and dipeptides derivatives were prepared by the reaction of coumarin-3-carboxylic acid with amino acid methyl esters following N,N-dicyclohexylcarbodiimide (DCC) and 1-hydroxy-benzotriazole (HOBt), as coupling reagents. Both compounds **4k** (Tyr) and **6c** (β-Ala-L-Met) exhibited potent cytotoxic activities against MCF-7 cells with IC_50_ values of 4.98 and 5.85 µM, respectively, causing cell death by 97.82 and 97.35%, respectively. Validating the molecular docking results, they exhibited promising VEGFR-2 inhibition with IC_50_ values of 23.6 and 34.2 µM, compared to Sorafenib (30 µM) and topoisomerase-II inhibition with IC_50_ values of 4.1 and 8.6 µM compared to Doxorubicin (9.65 µM). Accordingly, compounds **4k** and **6c** were singled out as promising therapeutic candidates for the research and development of new anticancer medicines due to their dual inhibitory effect.

## Data Availability

Data are contained within the article and Appendix A.

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
