# Peer review of "Facile Synthesis of Some Coumarin Derivatives and Their Cytotoxicity through VEGFR2 and Topoisomerase II Inhibition"

_molecules, 2022, doi:10.3390/molecules27238279_

Round 1

Author Response

Title: Convenient Synthesis of some Coumarin-based Amino Acid and Dipeptide Derivatives as cytotoxic agents through VEGFR2 and Topoisomerase II inhibition with In Silico approaches

It is my pleasure to review the manuscript submitted by Gomaa et al. (molecules-2021190). In this article, they reported coumarin-based peptide derivatives. Natural product-related synthetic compounds greatly contribute to the development of novel bioactive compounds. In this respect, the present study is important.

However, this study simply describes the coupling of coumarin-3-carboxylic acid with amino acids followed by molecular docking, ADMET, and cytotoxic assessment against MCF-7 and HepG2 cancer cell lines. I found the presentation style and writing very weak. In addition, authors have synthesized a good number of coumarin-based compounds but they did not add any spectrum. They should add spectra as a supplementary file, which will support/validate the research work.

I prefer authors read the manuscript carefully and improve substantially.

Response: Thanks for your comment. Kindly be informed that we prepared supplemenrty file for all the characterization analyses  

Some of the flaws are mentioned below:

  • Title: The title seems lengthy. In addition, authors have synthesized mono- and di-peptides from coumarin-3-carboxylic acid. Hence, ‘Coumarin-based Amino Acid and Dipeptide Derivatives’ is confusing. If possible, make the title

Response: We changed it

[2]  Abstract and keywords:

-Please rewrite lines 19-21 (just ‘N,N-dicyclohexylcarbodiimide (DCC)’ method is sufficient).

Response: We changed it

  • Insert alternate keywords for ‘In silico studies’

Response: Done as suggested

  • Introduction: No need to use an unnecessary capital letters in the middle of a sentence. For example, 7- Hydroxy-4-methylcoumarin (4-Methylumbelliferone) is not a brand name and it must be 7- hydroxy-4- methylcoumarin (4-methylumbelliferone).

Response: Done as suggested

  • Please check – ‘C-Jun NH2 terminal kinase’ (it should be ‘c-Jun NH2-terminal kinase’ or ‘c-Jun N-terminal kinase’.

Response: Done as suggested

  • Need to improve Table 2 and Figure

Response: Done as suggested. Figure 4 was updated, but Table 2 seem to be good at present form.

  • Figure 5 must be improved (legends and symbols).

 Response: Done as suggested.

  • Results and discussion: In this section, chemistry writing seems to be weak. For example, ‘coumarin 3-carboxylic acid’ must be ‘coumarin-3-carboxylic acid’ and so

Response: Done as suggested.

  • In section 2.1, please improve the writing. ‘In view of these facts and in continuation of our efforts in synthesizing various bioactive molecules,34 we have…..’ – what are these facts? Also, avoid using ‘we’.

Response: Done as suggested

  • In Scheme 3, please check reagents in 2nd step (it should be n2 in …(CH2)n1…). Authors may add yieldsin Schemes.

Response: Done as suggested

Figure 2 is not a spectrum. Please add related spectra of compounds.

Response Done as suggested and the spectra of compound will attached with the submitted data

  • Figure 3a should be rechecked (2 figures are merged).

Response: Done as suggested

[5] Experimental:

  • Please change ‘3.1. Chemistry’ to ‘3.1. Synthesis’. Many sentences are incomplete and hence meaningless (e.g. TMS (0.00 ppm) as internal standard.’; ‘From glycine methyl ester hydrochloride.’;

Response: Done as suggested

  • Please change ‘1H NMR spectrum, (300 MHz, CDCl3), δ, ppm (J, Hz):’ to ‘1H NMR (300 MHz, CDCl3): δppm’ throughout the manuscript. Change ‘9.22 (brs, 1H, NH);’ to ‘9.22 (br s, 1H, NH);’ ; ‘m.p.101-103o’ to ‘mp 101-103 °C.’; and so on.

Response: Done as suggested

  • In section 3.2.1., please mention grid box size. Please recheck – ‘In addition, hydrogens were incorporated into the protein during the manufacturing process. Amino acid residues with missing atoms and those with alternate positions for some of their atoms were optimized.’ Also, add some references

after    –‘AutoDock   Vina    was    used    for    molecular    docking.’.     (authors    may    consider- https://doi.org/10.3390/molecules27134125)

  • Please add more references in section -3.2.2. ADME pharmacokinetics (authors may consider - https://doi.org/10.1007/s11094-022-02687-y).

Response: This part was revised, grid-box dimension was added, and reference was added

Conclusion: Please add a conclusion concisely.

Response: Conclusion was added as suggested.

Reviewer 2 Report

I have some suggestions:

N. Duangdee, W. Mahavorasirikul and S. Prateeptongkum, Journal of Chemical Sciences, 2020, 132, 66.     This reference does not exist line 488

The title of the articles is not specified in the reference notes, I think they should be

The conclusions are not found, they should be discussed, especially since the authors must make a delimitation regarding the reaction intermediates and the new compounds obtained. It is not very clearly mentioned which of the synthesized compounds are new.

Authors should specify the contributions of each.

Author Response

Reviewer 2 comments  

  1. Duangdee, W. Mahavorasirikul and S. Prateeptongkum, Journal of Chemical Sciences, 2020, 132, 66.     This reference does not exist line 488

Response: Kindly be notified that reference [20] is present in the introduction, Page 2, as yellow highlighted in the manuscript. 

The title of the articles is not specified in the reference notes, I think they should be

Response: Thanks, we already all titles in the reference list.

The conclusions are not found, they should be discussed, especially since the authors must make a delimitation regarding the reaction intermediates and the new compounds obtained. It is not very clearly mentioned which of the synthesized compounds are new.

Response: Conclusion was added 

Authors should specify the contributions of each.

Response: New paragraph with author contribution was added following molecules format.

Round 2

Reviewer 1 Report

In the revised manuscript molecules-2021190-v2, the authors have improved in many sections. However, some more corrections/modifications are needed to be considered for a better quality of the manuscript. Authors must improve writing in the experimental section (please check any related good articles published in molecules). Some of them are as follows:

- In the Title please write ‘coumarin’ instead of ‘Coumarin’

- Many typos must be improved. In this respect, authors may check any good articles from Molecules.

- In line 83, it should be ‘coumarin-3-carboxylic’ instead of ‘coumarin- 3-carboxylic’. Similarly, check line 81, and so on.

- Please change ‘gave coumarin 3-carboxylic acid (3), Scheme 1. [32]’ to ‘gave coumarin-3-carboxylic acid (3, Scheme 1) [32].’ Please follow this style throughout the manuscript.

- Make a single paragraph for lines 98-102.

- Authors have not corrected the experimental section properly.

- In line 187, write as – Tetramethylsilane (TMS, 0.00 ppm) is used internal standard.

- In Scheme 2, please remove colors as it has no special meaning (in the present form).

- In line 118, please write as – Figure 2. Selected 1H NMR δ values of compounds 4e and 7a.

- In lines 199-344, check grammar. Also, write °C (not oC), and 1H NMR (not 1H NMR) throughout the manuscript.

- Insert °C in line 339.

I highly request authors check the manuscript carefully and compare it with any published good articles (otherwise it will be difficult to recommend for publication).

Author Response

However, some more corrections/modifications are needed to be considered for a better quality of the manuscript. Authors must improve writing in the experimental section (please check any related good articles published in molecules). Some of them are as follows:

- In the Title please write ‘coumarin’ instead of ‘Coumarin’

Response

done

- Many typos must be improved. In this respect, authors may check any good articles from Molecules.

- In line 83, it should be ‘coumarin-3-carboxylic’ instead of ‘coumarin- 3-carboxylic’. Similarly, check line 81, and so on.

Response

done

- Please change ‘gave coumarin 3-carboxylic acid (3), Scheme 1. [32]’ to ‘gave coumarin-3-carboxylic acid (3, Scheme 1) [32].’ Please follow this style throughout the manuscript

Response

done

- Make a single paragraph for lines 98-102.

Response

done

- Authors have not corrected the experimental section properly.

- In line 187, write as – Tetramethylsilane (TMS, 0.00 ppm) is used internal standard.

Response

done

- In Scheme 2, please remove colors as it has no special meaning (in the present form(.

Response

done

- In line 118, please write as – Figure 2. Selected 1H NMR δ values of compounds 4e and 7a.

Response

done

- In lines 199-344, check grammar. Also, write °C (not oC), and 1H NMR (not 1H NMR) throughout the manuscript.

Response

done

- Insert °C in line 339.

Response

done